# Eternal Youth: A Comprehensive Exploration of Gene, Cellular, and Pharmacological Anti-Aging Strategies

**DOI:** 10.3390/ijms25010643

**Published:** 2024-01-04

**Authors:** Kristina V. Kitaeva, Valeriya V. Solovyeva, Nataliya L. Blatt, Albert A. Rizvanov

**Affiliations:** 1Institute of Fundamental Medicine and Biology, Kazan Federal University, 420008 Kazan, Russia; krvkitaeva@kpfu.ru (K.V.K.); vavsoloveva@kpfu.ru (V.V.S.); nlblatt@kpfu.ru (N.L.B.); 2Division of Medical and Biological Sciences, Tatarstan Academy of Sciences, 420111 Kazan, Russia

**Keywords:** aging, senolytic, senescent cells, AAV, anti-aging therapy, cell therapy, gene therapy

## Abstract

The improvement of human living conditions has led to an increase in average life expectancy, creating a new social and medical problem—aging, which diminishes the overall quality of human life. The aging process of the body begins with the activation of effector signaling pathways of aging in cells, resulting in the loss of their normal functions and deleterious effects on the microenvironment. This, in turn, leads to chronic inflammation and similar transformations in neighboring cells. The cumulative retention of these senescent cells over a prolonged period results in the deterioration of tissues and organs, ultimately leading to a reduced quality of life and an elevated risk of mortality. Among the most promising methods for addressing aging and age-related illnesses are pharmacological, genetic, and cellular therapies. Elevating the activity of aging-suppressing genes, employing specific groups of native and genetically modified cells, and utilizing senolytic medications may offer the potential to delay aging and age-related ailments over the long term. This review explores strategies and advancements in the field of anti-aging therapies currently under investigation, with a particular emphasis on gene therapy involving adeno-associated vectors and cell-based therapeutic approaches.

## 1. Introduction

Increasing life expectancy is a global trend linked primarily to better sanitation, medical advances, improved nutrition, and safer living conditions fostered by social interaction. Population aging poses a significant social challenge, particularly in developed countries where the proportion of the elderly population continues to rise annually. Anti-aging therapy, also known as rejuvenation therapy, encompasses a spectrum of interventions designed to slow, reverse, or mitigate the health effects of aging. The significance of anti-aging therapy lies in its capacity to enhance human health, improve quality of life, and extend the period of healthy aging. By targeting the fundamental mechanisms of aging, anti-aging therapy aims to postpone the onset of age-related diseases and facilitate improved tissue regeneration along with the normalization of physiological functions [1].

The idea, initially proposed by Peter Medawar and later encapsulated in the 1957 hypothesis of “antagonistic pleiotropy” by American evolutionist George Williams, remains widely accepted and serves as the primary explanation for the evolution of aging today [2]. From the standpoint of classical medicine and common logic, aging is a degenerative progressive process that results in tissue dysfunction and death [3]. Signs of aging, manifesting at the cellular and molecular levels, are universal across organisms and include decreased genome stability, telomere depletion, mitochondrial dysfunction, epigenetic noise, and stem cell depletion and dysfunction [4].

Several factors are known to be involved in the aging process, activating cellular withering pathways, and genetic variations, as well as epigenetic modifications, can influence the rate of aging [5]. Certain genes are associated with longevity, while others may increase susceptibility to age-related diseases. Epigenetic changes, such as DNA methylation and histone modifications, can impact gene expression patterns and contribute to aging [6]. Aging initiates at the molecular level, manifests at the cellular level, and results in the aging of entire organ systems. For instance, aging of the adaptive immune system leads to the development of a chronic pro-inflammatory state characterized by sluggish inflammation, stemming from chronic activation of the innate immune system, termed “inflammaging” [7]. The immune system undergoes age-related changes, resulting in impaired immune responses and heightened susceptibility to infections, autoimmune diseases, and age-related chronic diseases [8,9]. Elevated levels of age-related proinflammatory markers are known to be detected in most older adults, even in the absence of risk factors and clinical activity [10]. During cellular senescence, L1 retrotransposable elements (also known as long interspersed nuclear element (LINE)-1) become transcriptionally derepressed and activate type I interferon (IFN)-I secretion. High IFN-I secretion is a late aging phenotype contributing to the maintenance of the secretory phenotype associated with aging [11].

Aging has been demonstrated to contribute to the development of several autoimmune diseases, including ANCA-associated vasculitis [12]. In aging and Alzheimer’s disease (AD), impaired meningeal lymphatic outflow promotes the accumulation of toxic misfolded proteins in the central nervous system (CNS). Consequently, therapeutic neutralization of IFNγ has been shown to alleviate age-related impairment of meningeal lymphatic function [13]. It is also known that the senescence of inflammatory tumor-associated fibroblasts (iCAF) in the tumor microenvironment leads to increased resistance of tumor cells to chemoradiation therapy. Interleukin 1 (IL-1) inhibition or senolytic therapy has been shown to prevent iCAF senescence and increase the sensitivity of mice to radiation. Additionally, lower serum levels of IL-1 receptor antagonist in patients with colorectal cancer correlate with a poor prognosis [14]. Studies have demonstrated that intra-articular injection of an IL-17-neutralizing antibody in a model of post-traumatic arthritis reduces joint degeneration and decreases the expression of the aging marker protein cyclin-dependent kinase 1A (CDKN-1A) inhibitor, also known as p21 [15]. Systemic inflammation, identified as one of the main but not the only markers of aging, is found in all organ systems, resulting from pathological changes at the cellular level [16].

Under physiological conditions, senescent cells can be eliminated by the immune system, promoting processes such as tumor suppression, embryogenesis, differentiation, and wound healing [17]. For instance, targeted induction of cellular senescence in malignant neoplasms is currently under active investigation: controlled senescence followed by apoptosis may be a crucial strategy for eliminating uncontrolled tumor cells from tissues [18,19,20]. Some of the most promising approaches to rejuvenate the body involve gene and cell therapy, combined with pharmacological intervention. These approaches aim to rejuvenate senescent cells, eliminate non-functional senescent cells, and block signaling pathways involved in cellular aging. This review will primarily focus on gene therapy approaches utilizing recombinant adeno-associated virus (AAV), as well as pharmacotherapy and cell-based interventions.

## 2. Cellular and Organ-Specific Aging

Cellular senescence was first reported and described as early as Leonard Hayflick and Paul Moorhead in 1961, and it was understood as a state of irreversible growth arrest occurring in response to telomere shortening [21]. Telomeres are known to consist of 10 to 15 base pairs of tandem hexanucleotide repeats (TTAGGG) of DNA located at the ends of linear chromosomes [22]. They serve as protective caps on the ends of chromosomes, shortening with each cell division, and telomere shortening is considered a sign of aging [23,24]. When a critical shortening of telomeric repeats is reached, the cell enters a state of sustained proliferative arrest, reaching what is termed replicative senescence. Proliferation arrest is a protective mechanism that, in the case of shortening or loss of telomeres at the ends of chromosomes, prevents unplanned DNA repair reactions leading to chromosomal instability and, consequently, malignant degeneration of the cell [25].

Critical telomere shortening as well as mitochondrial dysfunction and genotoxic stress lead to double-stranded DNA breaks, which activate the p53/p21 oncosuppressor signaling pathway. High levels of p21 inhibit the kinase activity of cyclin D/cyclin-dependent kinase (CDK)-4/6 complexes, resulting in the inhibition of cell cycle progression, whereas low levels of p21 act as an assembly factor of the cyclin D/CDK4/6 complex and promote its activation leading to cell cycle progression [26]. In addition to telomere shortening, there are other mechanisms for induction of cell aging, such as through activation of oncogenes such as *BRAF* (V600E mutation) or *RAS*, which is called oncogene-induced senescence, or inactivation of an oncosuppressor gene such as *PTEN* [27,28]. This form of cellular senescence was discovered in 1997. It was found that transfection of oncogenic *HRAS* (V12 mutation) into mouse, rat, and human embryonic fibroblasts resulted in the arrest of cell proliferative activity [28,29].

The oncosuppressor gene p16, an inhibitor of cyclin-dependent kinase 4A, is specifically activated in senescent cells and suppresses CDK4/6, thereby inducing senescence-associated cell cycle arrest in the G0/1 phase, which depends on the dephosphorylation of retinoblastoma Rb family proteins (Rb, p107 and p130), which are known targets of CDK4/6 [30]. P16 is one of the markers of aging and its upregulation is confirmed in many senescent cells and tissues [31]. It is known that deletion of p16-positive senescent cells delays the onset and progression of age-related pathologies in mice in vivo and prolongs the lifespan of prematurely and naturally aging mice [32,33]. Thus, prolonged overexpression of any of these four critical components (p53, pRB, p16I^NK4A^ and p21^WAF1/CIP1^) is sufficient to induce senescence [34].

In addition to cell cycle arrest, senescent cells exhibit increased β-galactosidase activity (referred to as senescence-associated β-galactosidase, SA-β-gal is one of the widely used markers of aged cells) at pH 6.0, activation of signaling pathways, and secretion of various growth factors, chemokines, and cytokines [35]. The activation of senescence-associated secretory phenotype (SASP) genes and the secretion of factors such as IL-6, IL-8, monocyte chemotactic protein-1 (MCP-1), vascular endothelial growth factor (VEGF), and transforming growth factor-β (TGF-β) have been demonstrated to mediate epithelial cell proliferation and play a role in the progression and development of various carcinomas [36]. Increased IL-8 secretion by senescent fibroblasts has been shown to stimulate the invasion and metastasis of a breast cancer cell line in an in vitro cell culture [37].

Another unique phenotype of senescent cells is the formation of senescence-associated heterochromatin foci (SAHF), which are specialized regions of facultative heterochromatin that reduce the expression of genes promoting proliferation [38]. Epigenetic changes, such as DNA methylation changes, histone modifications, and the expression of non-coding RNAs, can influence gene expression patterns without altering the underlying DNA sequence. These epigenetic modifications can impact cellular function, aging processes, and susceptibility to age-related diseases [39].

Acute aging, associated with a programmed cellular strategy in response to various stressors, is believed to be involved in homeostatic biological processes. However, chronic aging is linked to prolonged exposure to cell-damaging stressors, resulting in a nonviable and burdensome cell cycle. This places an enormous load on the reparative and regulatory pathways required to maintain cell integrity. Unlike acute aging, chronic aging is not a programmed response; it is persistent and is believed to play a deleterious physiological role, contributing to aging (senility) and age-related diseases [40] (Figure 1).

As inferred from the preceding section, the initiation of aging at the cellular level is an established concept. Furthermore, there exist studies that have delineated aging clocks for specific organs. For instance, algorithms have been devised based on such studies to evaluate the physiological functions of organs such as the kidneys and lungs. These algorithms take into account the interplay between chronological and biological ages to assess the functional status of these organs [41,42]. It has been demonstrated that accelerated aging is characteristic of several organ systems in conditions such as ischemic heart disease, hypertensive disorders, diabetes, osteoarthritis, and cancer. Moreover, the biological age of an organ has the capacity to predict the risk of mortality [43]. Another question pertains to identifying the organ-specific aging-related proteins that act as causal factors in the aging process. This consideration is particularly relevant given that numerous proteins, such as KLOTHO, UMOD, MYL7, CPLX1, CPLX2, and NRXN3, exhibit genetic associations with diseases specific to respective organs or are confirmed therapeutic targets. This circumstance leads to the conjecture of a potential causal role for these proteins in the aging process [44]. During the RNA sequencing of 17 organs from C57BL/6JN mice of varying ages, it was observed that genes involved in the regulation of the extracellular matrix, binding of unfolded proteins, mitochondrial function, as well as inflammatory and immune responses, are uniformly expressed across different tissues, differing only in the age of onset and amplitude of expression. Notably, a high correlation was demonstrated for vascular cell adhesion molecule-1 (Vcam1) in the kidneys and fibroblast growth factor 10 (Fgf10) in the spleen, as well as for glial fibrillary acidic protein (Gfap) and brain protein. It is also noteworthy that the level of the plasma B-cell marker immunoglobulin J chain (Igj/Jchain) exhibits a persistent increase throughout life in 11 out of 17 organs. The circadian clock genes *Bhlhe40/41*, *Arntl*, *Npas2*, *Per3*, *Ciart*, and *Dbp* also emerge as among the top differentially expressed genes, suggesting potential involvement in metabolic and inflammatory disorders, as well as a reduction in lifespan associated with circadian rhythm disruption [45]. In another study, it was demonstrated that aged mice exhibited an elevation in the plasma level of von Willebrand factor (vWF). Concurrently, mRNA levels of vWF and cellular protein were significantly increased in the brain, lungs, and liver, but not in the kidneys and heart, of aged mice [46]. It has been discovered that aging can impact chemotaxis, as reflected by levels of MCP-1, NAP-3, and eotaxin, influencing subsequent immune cell infiltration in an organ-specific manner. Notably, the liver and spleen exhibit pronounced effects, while other organs are comparatively less affected [47]. It is well-established that the expression of p16 increases in aged cells. For instance, in a murine model, predominant mRNA and protein expression of p16 were demonstrated in hepatic endothelial cells compared to non-endothelial cells in aged mice, suggesting a functional role specifically within the liver endothelium of aged subjects. Moreover, it has been shown that the dynamic expression of p16 may be implicated not only in aging processes but also in ontogenetic and physiological processes [48]. Furthermore, it has been demonstrated that p16INK4a-positive lung fibroblasts contribute to the growth of respiratory tract stem cell organoids, suggesting their potential role in supporting epithelial regeneration. Interestingly, the removal of p16INK4a-positive cells or the blockade of p16INK4a expression in mice—achieved through senolytic agents or Cre-mediated deletion from fibroblasts—resulted in a reduction in the growth of respiratory tract stem cells induced by naphthalene-induced tissue damage and disrupted tissue repair, leading to enhanced fibrosis [49].

However, in the context of intestinal organoids as a model for tissue regeneration, it has been demonstrated that senescence-associated secretory phenotype (SASP) factors, particularly the secreted N-terminal domain of Ptk7, released by aging fibroblasts, dysregulate the activity and differentiation of stem cells, ultimately disrupting crypt formation [50]. In model systems of oxidative-stress-induced aging in vitro and ex vivo in fibroblasts of the colon, it has been demonstrated that aging fibroblasts secrete GDF15, which promotes the proliferation, migration, and invasion of cells in colorectal adenoma and colorectal cancer cell lines, as well as primary colon organoids, through the MAPK and PI3K signaling pathways. This observation suggests a potential similarity in the pathways involved in cancer development and aging [51]. The organ specificity of aging is supported by epigenetic data, indicating that age-related changes in DNA methylation are also highly tissue-specific, with the exception of CpG sites in the ELOVL2 promoter. Age-related enhancement of DNA methylation (gain-aDMPs) accumulates on CpG islands and their flanking regions, which are associated with the repressive PRC2 EZH2 component. These modifications may serve as informative markers for biological age [52]. In general, the organ specificity of aging constitutes a complex scenario wherein various organs may respond differently to the aging process. This underscores the significance of considering this organ specificity in the investigation of aging mechanisms and the development of strategies to maintain health in the elderly.

## 3. The Use of Senolytic Agents

Senolytic drugs are designed to eliminate senescent cells—cells that can no longer perform their functions but persist, exerting a negative impact on surrounding healthy cells. Such cells can contribute to the development of various age-related diseases and the associated pathological aging of the body. Research on senolytic drugs is focused on identifying substances capable of inducing apoptosis (programmed cell death) in senescent cells, thereby preventing their detrimental effects on healthy cells and the body as a whole. Over 46 potentially senolytic compounds targeting senescent cell antiapoptotic pathways (SCAP) have been identified. These include the SRC tyrosine kinase inhibitor dasatinib, which has been approved and widely used since 2006 with a relatively good safety profile, as well as the natural flavonoids quercetin and fisetin (refer to Table 1) [53]. These drugs belong to the first generation of senolytics, and their action is directed at various molecular targets and signaling pathways, such as tyrosine kinase receptors, growth factors, ephrin B1, SRC kinases, the phosphoinositide-3-kinase (PI3K)/protein kinase B (AKT) signaling pathway, heat shock protein-90 (HSP-90), BCL-2 family members (regulators of apoptosis), caspases, and p53 [54].

One of the best-known senolytic drugs is dasatinib, originally developed for the treatment of blood cancers but found to be effective in killing senescent cells. Type 2 diabetes mellitus (DM2) is recognized as an age-related disease, and insulin resistance accelerates β-cell aging [55]. We showed that senolysis of p21^high^ cells in human adipose ex vivo xenografted immunodeficient mice using a dasatinib + quercetin cocktail reduces insulin resistance [56]. Overall, this drug combination is a popular tool for increasing the clearance of senescent cells in tissues [57,58,59,60].

In addition to dasatinib and quercetin, other drugs, some of which are also antitumor agents, have great potential for anti-aging therapy. For example, rapamycin (sirolimus) and its analogs (everolimus, temsirolimus, and deforolimus) bind to the cytosolic protein FKBP12 and thus inhibit the mammalian target of rapamycin complex 1 (mTORC1), reducing the incidence of malignant neoplasms in kidney transplant patients [61]. mTOR is a serine/threonine kinase known to play a central role in the regulation of cellular metabolism and growth by phosphorylating various substrates in response to growth factors, stress, nutrient availability, and other stimuli. Therefore, targeting mTOR signaling pathways is one of the promising methods for slowing aging [62].

The addition of a FOXO4-related peptide to senescent fibroblasts of the human IMR-90 cell line has been shown to reduce their viability by more than 10-fold compared to non-senescent IMR-90 fibroblasts or other cell types. The mechanism of action of the FOXO4-related peptide is based on preventing the binding of the transcription factors FOXO4 and p53 in the cell nucleus, ultimately leading to the release of the p53 protein into the cytosol and triggering caspase-dependent apoptosis of senescent cells [63].

HSP90 is a cytoplasmic protein that prevents proteasomal degradation of AKT and promotes lung tumor cell survival. Thus, HSP90 inhibitors are actively being studied as a therapeutic agent for the treatment of malignant neoplasms [64]. In addition to their antitumor activity, drugs that inhibit HSP90, such as geldanamycin, have senolytic effects against senescent cells [65].

In some cases, senolytic compounds targeting a single SCAP node, such as inhibitors of the BCL-2 signaling pathway (ABT-263), N (A1331852), or A1155463, tend to induce apoptosis in a limited range of senescent cell types; however, their administration may cause toxic effects on the body [66].

Second-generation senolytics include agents based on lysosomal and SA-β-gal-activated drugs, nanoparticles, sodium–potassium pump gradient-dependent apoptosis activation (Na^+^/K^+^-ATPase), SASP inhibitor drugs, and induction of clearance of senescent cells by antibody–drug conjugates, vaccination, and cell therapy based on chimeric antigen receptor synthesizing T cells (CAR T cells), which will also be mentioned in the cell therapy section [54].

However, for the widespread use of these drugs in clinical practice, further research is required to find the most effective combinations and verify their safety for humans.

**Table 1 ijms-25-00643-t001:** List of the most common pharmacologic senolytics.

Name	Mechanism of Action	Model Used	Effect	Reference
A-1155463	Selective inhibitor of BCL-XL	*SCID-Beige* mice with xenograft of BCL-XL-dependent lung carcinoma cells of the NCI-H146 lung carcinoma cell line	Causes reversible thrombocytopenia in mice and inhibits small-cell lung cancer xenograft growth in vivo after repeated administration	[67]
		IMR-90 cell line, primary human preadipocytes, HUVECs	Induces apoptosis of senescent HUVEC and IMR-90 cells, but not preadipocytes	[66]
		Gastric cancer cell lines (23132/87, SNU216, NCI-N87, MKN1, AGS, HGC27, SNU719). Human multiple myeloma cell lines (MM1S, KMS12PE, KMS12BM)	Has a cytostatic effect on tumor cells	[68]
A-1331852	Selective inhibitor of BCL-XL	Mouse model of glioblastoma multiforme, U251 and SNB-19 cells	Promoted the destruction of U251 cells	[69]
Cardiac glycosides	Inducers of apoptosis through the Na^+^/K^+^-ATPase pump	PDX-immunodeficient *NMRInu/nu* mice with xenografts of A549 (lung adenocarcinoma) and IMR-90 (normal human lung cells) cells	In vivo inhibition of xenografted tumors in mice after treatment	[70]
Curcumin	Decreased expression of the opioid nociceptin receptor gene (*OPRL1*)	Human neuroglia cell line T98G	Reduces *OPRL1* gene expression associated with pain syndromes	[71]
	Inhibition of the mitogen-activated protein kinase (MAPK)/transcriptional nuclear factor κB (NF-κB) signaling pathway	C57BL/6 mice and primary hepatocytes isolated from the livers of C57BL/6 mice	Inhibits the MAPK signaling pathway in the liver in old mice and p38 in old mice with diet-induced obesity. Improves insulin homeostasis and reduces body weight in old mice	[72]
Dasatinib + quercetin	Suppression of SRC family kinase inhibitors	DU-145 and LNCaP prostate cancer cells	Inhibits the cell adhesion, migration, and invasion of prostate cancer cells at low nanomolar concentrations	[73]
	Decreased mRNA levels of genes encoding proinflammatory cytokines: IL-1β, tumor necrosis factor α (TNF-α) and chemokine (C-X-C motif) ligand 8b.1 (CXCL-8b.1)	*Danio rerio* fish larvae deficient in the serine protease inhibitor protein Spint1a	Has an anti-inflammatory effect and reduces chronic inflammation	[7]
Fisetin	Blocks the signaling pathway PI3K/AKT/mTOR/p16^INK4a^	*Ercc1*^−/∆^ mice (human progeroid syndrome model) and aged wild-type mice, human fibroblasts (IMR-90)	Tissue-specifically reduces cellular senescence in mouse adipose tissue and human cells	[74]
FOXO4-related peptide	Blocks the interaction of transcription factor FOXO4 with p53, leading to apoptosis	Human chondrocytes of early and late passages	Removes (eliminates) senescent cells in the late passage chondrocyte population in vitro	[75]
Geldanamycin	HSP90 inhibitor	Primary embryonic fibroblasts from *Ercc1*^−/−^/mesenchymal stem cells (MSCs) from the bone marrow of *Ercc1*^−/Δ^ and *Ercc1*^−/∆^ mice	Prolongs the lifespan of mice with a progeria model, delays the onset of several age-related symptoms, and reduces the expression of p16^INK4a^	[76]
Luteolin	Inhibitor of mTOR signaling pathway	Human bladder cancer cell lines T24, 5637 (with p53 mutation), and RT-4 and rat bladder cancer cell line BC31 (with p53 mutation) in vitro/rats with bladder cancer model	Inhibits cell survival and induces cell cycle arrest in the G2/M phase; p21 activation in bladder cancer cells	[77]
Navitoclax (previously ABT263)	BCL-2 inhibitor	Human skin xenograft in immunodeficient mice	Causes selective elimination of senescent dermal fibroblasts	[78]
Nutlin-3a	E3 ubiquitin ligase inhibitor MDM2/p53	A chemically induced aging mouse model, an Alu-induced geographic retinal atrophy model, and aged mice	Has a senolytic effect; reduces levels of aging markers, SASP components, and ocular pigment deposits	[79]
Piperlongumin	Inhibitor of extracellular signal-regulated kinase (ERK) 1/2	Senescent human fibroblasts of WI-38 lineage	Shows moderate selectivity in reducing the viability of ionizing-radiation-induced senescent fibroblasts of the WI-38 lineage	[80]
Rapamycin	Inhibitor of mTOR signaling pathway	Nrf2-KO fibroblasts (nuclear factor Nrf2 knockout) in vivo, Nrf2-KO mice in vitro	Increases Nrf2 levels, which activates autophagy, and reduces induction of cellular senescence in vitro. In mice, Nrf2-KO reduces the concentration of proinflammatory cytokines in serum and adipose tissue in vivo	[81]
Resorcin	HSP90 inhibitor	Primary embryonic fibroblasts from *Ercc1*^−/−^ and *Ercc1*^−/∆^ mice	Reduces the number of senescent mouse embryonic cells	[76]
Tanespimycin (17-AAG)	HSP90 inhibitor	An isogenic model of BAX knockout in human colon carcinoma cell line HCT116 in vitro and in tumor xenografts in vivo.	Causes a cytostatic antiproliferative effect on tumor cells through inhibition of oncogenes	[82]
Alvespimycin (17-DMAG)	HSP90 inhibitor	Primary embryonic fibroblasts from *Ercc1*^−/−^ and *Ercc1*^−/∆^ mice	Prolongs the lifespan of Ercc1^−/∆^ mice, delays the onset of several age-related symptoms, and reduces p16^INK4a^ expression	[76]

## 4. Cell and Genetically Modified Cell-Based Therapy

Cell therapy is considered one of the promising approaches for correcting age-related diseases, involving the use of the regenerative and immunomodulatory properties of cells to restore tissues’ functions and improve overall health. However, in the context of anti-aging cell therapy, there are nuances to consider when choosing a treatment strategy. It is known that the regenerative capacity of tissues decreases with age. For example, studies have shown that when using cells of bone marrow origin, young recipients exhibit statistically better skin healing efficiency than older recipients [83]. This property is manifested at the organ level; for instance, in adult recipients, kidneys from young donors show an advantage in engraftment and a lower risk of rejection compared to kidneys from older donors [84]. Moreover, it has been shown that the organs of elderly donors had a negative effect on the recipients, accelerating the aging of their bodies [85].

It was shown that human skin xenografts from elderly donors transplanted into young immunodeficient mice were morphologically rejuvenated within the first month after transplantation. However, interestingly, during the following year, skin rejuvenation leveled off, and the transplanted sections returned to their condition before being transplanted into mice [86]. The data from transplantation studies are consistent and confirmed by in vitro studies. Co-culture of epithelial progenitor cells isolated from aged mice with MSCs or with membrane vesicles isolated from the MSCs of young mice resulted in the rejuvenation of “old” epithelial progenitor cells [87]. It was shown that cardiosphere-derived cells (CDCs), which are cardiac progenitor cells isolated from the hearts of neonatal rats, reproduced the juvenile pattern of gene expression when injected into the hearts of old animals. Telomeres in heart cells were also longer in animals after CDC transplantation [88]. Injection of cerebrospinal fluid from young mice directly into the brain induced oligodendrocyte proliferation and consolidation of long-term memory in old mice [89]. In view of these findings, using cells isolated from young donors or placing senescent cells in a medium containing factors characteristic of a young organism may be one tool for tissue rejuvenation.

The use of stem cells demonstrates effectiveness due to their ability to differentiate into various cell types, regenerate, or replace damaged tissues and organs [90,91]. Autologous MSCs are considered safe and may be effective for certain conditions [92]. For example, the inclusion of MSCs in standard therapy for the early phase of acute severe pancreatitis in patients of the middle-aged group (approximately 44 years old) allows purposeful and relatively quick intervention in abnormal homeostatic processes, inhibiting toxic phenomena, restoring immune response, and improving microcirculation [93]. Positive results were obtained using bone marrow cells multiplied in vitro and injected into the defect site along with biphasic calcium phosphate granules, inducing the formation of new bone. In this case, the volume of regenerated bone was sufficient for dental implant placement in patients aged 52–79 years, with satisfactory aesthetic and functional results and no reported side effects (NCT02751125) [94].

Moreover, MSCs and adipose-tissue stem cells offer an effective alternative to reduce or slow down the aging process of facial skin [95]. It was shown that natural MSC exosomes both had neuroprotective effects in the microglia culture line BV2 in vitro and exhibited anti-apoptotic and antioxidant effects in the brains of aging SAMP8 mice with an accelerated aging phenotype [96]. MSCs overexpressing sirtuin (SIRT) 3 improved rat cardiac function and increased VEGF-A levels and vascular density [97]. However, despite the versatility and safety of MSC use, questions regarding their efficacy for certain conditions, such as osteoarthritis, remain [98].

Another option for cell therapy is the use of stromal vascular fraction (SVF) and platelet-rich plasma (PRP). SVF can enhance angiogenesis and neovascularization in wound healing and urogenital and cardiovascular diseases [99]. Coronary microvascular levels of β1 adrenergic receptors are known to decrease with age, but this decrease in expression was restored by treatment with SVF. Intravenous administration of SVF to aged rats improved their coronary microvascular function [100]. The application of autologous microfragmented adipose tissue with SVF in patients with knee osteoarthritis increased glycosaminoglycan levels in hyaline cartilage in older patients (63+ years of age), resulting in decreased pain and improved motor ability [101].

Platelet-rich plasma (PRP) is plasma with a platelet content several times higher than that of blood plasma. It has been demonstrated that PRP stimulates the activity of glycolytic enzymes in fibroblasts, decreases the rate of oxygen consumption, and impacts certain types of mitochondrial respiration. Moreover, PRP activates SIRT1 expression, contributing to the rejuvenation of fibroblasts [102]. According to clinical data, intra-articular injection of PRP into patients with osteoarthritis did not result in adverse effects. However, due to the peculiarities of PRP preparation and significant variations in the ratio of cellular fractions in the final product, further studies and validation of this therapeutic method are required [103]. Another cell variant studied for its therapeutic potential is RCS-01. This preparation consists of 25 × 10^6^ cultured autologous cells derived from the nebulbar dermal sheath of a hair follicle in the anagenic state. RCS-01 injections were well tolerated, with no reported serious side effects. A single injection of RCS-01 led to a significant increase in the mRNA expression of *TGF-β1*, connective tissue growth factor (*CTGF*), type I collagen (*COL1*) *A1*, *COL1A2*, *COL3A1*, and lumican genes (NCT02391935) [104].

A major trend in cell therapy involves the use of reprogrammed and genetically modified cells. Cell reprogramming entails resetting the epigenetic state of cells, reverting them to a younger state, with the goal of reversing cellular aging and restoring function. For instance, the use of induced pluripotent stem cells (iPSCs) represents a promising therapy for age-related macular degeneration, one of the leading causes of irreversible visual impairment globally, characterized by degeneration of the retinal pigment epithelium. Several studies have proposed therapeutic options based on retinal pigment epithelial cells differentiated from allogeneic iPSCs, with successful trials conducted on various animal models, including allografts in macaque monkeys [105,106,107,108].

However, not all experiments with the introduction of iPSCs have been equally successful. Old rats injected with human neural progenitor cells derived from iPSCs at the site of chronic cervical spinal cord injury showed no improvement in behavioral tests. Additionally, high mortality rates were observed during behavioral training (41.2%), after injury (63.2%), and after cell injection (50%). Meanwhile, histological analysis revealed that the injected cells survived and remained at the transplant site, without inducing tumors, confirming their safety [109]. Modified tendon fibroblasts expressing angiogenic factors (placental growth factor (PlGF) and matrix metalloproteinase-9 (MMP9)) can repair the vasculature and reduce collagen deposition, enabling effective cell therapy in aged mice with dystrophy [110]. Genetically modified effector immune cells, such as CAR T cells targeting the urokinase-type plasminogen activator surface receptor (uPAR), induced during senescence, have shown high efficacy in removing senescent cells both in vitro and in vivo, as well as in restoring tissue homeostasis in mice with induced fibrosis [111]. Thus, one of the key aspects of implementing cell therapy lies in its potential role in the prevention of age-related diseases, such as cancer, heart diseases, and immune system dysfunction. Tissue regeneration and the maintenance of tissue health can play a crucial role in reducing the risk of developing these conditions. It is noteworthy that despite the promising prospects of cell therapy, this approach is still in the active research stage. Further investigations and developments are required for a comprehensive understanding of the capabilities and limitations of cell therapy in the context of combating aging. The safety and effectiveness of this approach remain pivotal aspects demanding careful scrutiny.

## 5. Possibilities for Gene Therapy Based on Recombinant Adeno-Associated Viruses

Recombinant AAVs are small, non-enveloped viruses, with the deletion of the rep and cap genes replaced by the insertion of the transgene of interest. The viral vectors contain 4.7 kb single-stranded genomic DNA with two palindromic GC-rich inverted end repeats at the ends of the chain [112]. Recombinant viruses are constructs comprising a promoter, genes of interest, and a terminator to enhance their suitability for clinical applications. As these recombinant AAVs cannot replicate, they provide a safe means for ensuring long-term transgene expression following a single infection. AAVs have emerged as efficient carriers for genetic modification due to their effective in vivo infectivity, non-pathogenic nature, broad tissue tropism, infrequent genomic integration, and ability to infect and persist in non-dividing cells [113,114]. Four decades of research have demonstrated that AAVs are among the safest and most effective vectors for delivering genes of interest to a diverse array of cell types in gene therapy applications [115,116]. Hereditary diseases are particularly attractive targets, and AAV vectors are well-established as prominent genetic therapies, including therapies for aging [113,117].

Several genes have been identified as potential targets for gene therapy aimed at enhancing longevity and health. These genes are often involved in signaling pathways that play a role in the regulation of cellular metabolism, oxidative stress, and inflammation, all of which are believed to contribute to the aging process. In this review, we will explore some of these genes, along with the possibilities of utilizing AAVs to address age-related diseases.

### 5.1. Sirtuins

Sirtuins constitute a family of proteins that play a crucial role in the regulation of cellular metabolism and stress response. Notably, Sirt1 overexpression, particularly in skeletal muscle, has been demonstrated to counteract the development of insulin resistance induced by a high-fat diet in mice. While the administration of AAV1-*Sirt1* alone did not suffice to prevent obesity and insulin resistance caused by a high-fat diet, it was observed that there was an increased expression of key genes related to β-oxidation, accompanied by elevated levels of phosphorylated AMP-activated protein kinase (AMPK). Furthermore, Sirt1 overexpression in skeletal muscle led to an augmentation of basal levels of AKT phosphorylation [118]. Additionally, intravitreal delivery of Sirt1 improved visual function in mice with diabetic retinopathy [119].

SIRT3 is broadly expressed in mitochondria-rich tissues with high metabolic demands, including brain, heart, kidney, muscle, and brown adipose tissue. Particularly in cells such as neurons, cardiomyocytes, and hepatocytes, SIRT3 plays a protective role against inflammation, oxidative stress, and senescence. For instance, gene expression and phenotype analysis of Sirt3 knockout mice revealed that reducing Sirt3 expression mediated persistent inflammation in the liver. Conversely, restoring Sirt3 expression in the liver effectively inhibited persistent liver inflammation and cardiovascular damage [120]. It was demonstrated that SIRT3 expression was suppressed in senescent human MSCs. CRISPR/Cas9-mediated inhibition of SIRT3 resulted in impaired nuclear integrity, loss of heterochromatin, and the accelerated aging of MSCs. SIRT3 was also shown to interact with nuclear envelope proteins and proteins associated with heterochromatin. SIRT3 deficiency led to the separation of laminin-associated genomic domains from the nuclear lamina, increased chromatin accessibility, and aberrant transcription of repetitive sequences [121]. In another study, receptor activator of NF-κB ligand (RANKL) was demonstrated to increase *Sirt3* mRNA and protein levels, thereby stimulating osteoclast formation and bone resorption, most likely through the stimulation of mitochondrial metabolism. In the absence of Sirt3, the mitochondrial function of osteoclasts decreases with age, resulting in reduced osteoclast function and the preservation of bone mass in mice [122]. Suppression of SIRT3 was found to further increase lactate dehydrogenase release, decrease ATP levels, suppress mitochondrial membrane potential, and increase oxidative stress in cardiomyocytes. SIRT3 deficiency further elevated the expression of necroptosis-related proteins, including receptor-interacting protein kinase 1 (RIPK-1), RIPK3, and cleaved caspase 3 (CASP-3), and increased the expression of inflammation-related genes, including NOD-like receptor family pyrin-domain-containing protein 3 (*NLRP3*), *CASP1*, *p20*, and *IL-1β* both in vitro and in vivo [123]. It was demonstrated that topical irisin treatment reduced alveolar bone loss and oxidative stress while increasing Sirt3 expression in periodontal tissues in experimentally induced rat models of diabetes and periodontitis. By culturing periodontal ligament cells (PDLCs) in vitro, it was found that irisin could partially restore suppressed cell viability, reduce intracellular oxidative stress and mitochondrial dysfunction, and restore impaired osteogenic and osteoclastogenic abilities in PDLCs upon exposure to high glucose and proinflammatory stimulation [124]. However, caution should be exercised in the use of a strategy to increase SIRT3 expression, ensuring tissue specificity to avoid exacerbating pathological effects.

*SIRT6* overexpression has been shown to preserve telomere integrity, delay cellular senescence, and reduce the expression of senescence-associated inflammatory cytokines and changes in vascular smooth muscle cell metabolism. SIRT6 promoted the proliferation and longevity of murine vascular smooth muscle cells and prevented metabolic changes associated with aging. However, the mutant variant SIRT6^H133Y^ (inactive deacetylase mutant variant of SIRT6) shortened the lifespan of both human and murine vascular smooth muscle cells [125]. Nevertheless, overexpression of SIRT6 has been demonstrated to induce carotid plaque hemorrhage by promoting angiogenesis in atherosclerotic plaques through increased expression of hypoxia-inducible factor 1α (HIF-1α) under hypoxic conditions, in conjunction with damage to newly formed vessels via reactive oxygen species under oxidative stress [126]. Thus, SIRT1 appears to be the most promising and safest target within the sirtuin family at this time.

### 5.2. Telomerase

Telomerase is an enzyme that plays a crucial role in maintaining the length of telomeres, protective caps at the ends of chromosomes. Telomere shortening is believed to contribute to the aging process. Researchers have explored the use of viral vectors to deliver telomerase-expressing genes into cells with the aim of slowing telomere shortening and promoting longevity. While there is considerable variation in telomere length between individuals, telomeres inevitably shorten with age and cell division [127]. A method for enhancing longevity using a cytomegalovirus vector encoding telomerase reverse transcriptase (*TERT*) and follistatin (*FST*) genes has been proposed and demonstrated to be highly effective in a mouse model of natural aging. When administered intranasally or by injection, gene therapy resulted in increased longevity (by more than 32%), improved glucose tolerance and physical endurance, and prevented weight loss and alopecia areata [128].

The application of gene therapy to express active human TERT (hTERT) in human cells holds the potential to treat numerous neurodegenerative diseases associated with aging, including Alzheimer’s disease (AD). Clinical trials involving this strategy are underway, such as one conducted by Libella Gene Therapeutics. This trial involves treatment with hTERT delivered via transduction using adeno-associated virus (AAV) (NCT04133454). The objective is to lengthen telomeres to prevent, delay, or even reverse the progression of AD. Telomere lengthening is anticipated to have a direct impact on cognitive function and the quality of life in patients with age-related neurodegenerative diseases such as AD. However, the use of hTERT is accompanied by the risk of malignant cell transformation. For instance, the hTERT/MDM2-FOXO3a-integrin β1 (ITGB1) signaling pathway is implicated in hTERT-stimulated gastric cancer invasion, suggesting that this signaling pathway may be a novel target for the prevention and treatment of gastric cancer metastasis [129,130].

### 5.3. The Aging Suppressor Gene Klotho

Klotho is an aging suppressor gene whose association with longevity has been demonstrated in animal models. It is known that the amount of Klotho decreases with age in humans and mice, while increasing its expression slows down or reverses age-related changes [131]. The *Klotho* gene encodes a type I membrane protein related to β-glucosidases. Reduced production of this protein has been observed in patients with chronic renal failure (CRF), and this may be one of the factors underlying the degenerative processes (e.g., arteriosclerosis, osteoporosis, and skin atrophy) observed in CRF. Additionally, mutations in this protein are associated with aging and loss of bone mass (GeneID: 9365). The effect of a deficiency of this protein was demonstrated in a mouse model with Klotho haplodeficiency, which caused impaired kidney function [132]. Klotho deficiency leads to heart failure in Klotho-hypomorphic mutant (KL^(−/−)^) mice [133]. In this study, we demonstrated that intra-articular delivery of the *α-Klotho* gene using plasmid DNA increased Klotho protein synthesis and delayed cartilage degradation in a mouse model of osteoarthritis [134,135].

The use of gene therapy to enhance Klotho activity in humans using AAV is actively being explored. For example, AAV-*Klotho* was injected into the bilateral hippocampus of rats with a model of temporal lobe epilepsy, and after 9 weeks, it was found that AAV-Klotho induced *Klotho* overexpression in the hippocampus, effectively improved cognitive impairment, and had a neuroprotective effect. Additionally, Klotho significantly increased glutathione peroxidase-4 and glutathione levels while suppressing reactive oxygen species levels in a rat model of temporal lobe epilepsy [136]. It was shown that the administration of AAV-*Klotho* to mice with a temporal lobe epilepsy model significantly attenuated hippocampal neuronal damage and cognitive impairment [137]. AAV-*mKlotho* (murine Klotho) prevented the progression of spontaneous hypertension, eliminated renal tubule atrophy and dilatation, and attenuated kidney damage in rats with spontaneous hypertension [138]. Neuroprotective and anti-inflammatory effects, as well as the restoration of the epigenetic landscape upon the administration of AAV9-*Klotho* to mice with a rapid aging model, were confirmed [139].

The administration of AAVs encoding a soluble form of the Klotho protein reduced arterial stiffness in aging mice, including by restoring the B-cell population and serum immunoglobulin G (IgG) levels, and attenuated aging-related vascular inflammation and arterial remodeling [140]. There is a known trial conducted on patients with mild to moderate dementia using AAV vectors encoding hTERT and KLOTHO, where the safety of the vectors and improvement of cognitive function were shown; however, official data on this study are not yet available [141]. Despite encouraging results after correcting Klotho protein levels using AAV vectors, it has not yet been possible to overcome the molecular entropy that inevitably occurs during the life of a cell and organism as a whole [142].

### 5.4. Fibroblast Growth Factor 21

Fibroblast growth factor 21 (FGF21) is a hormone that plays a crucial role in the regulation of glucose and lipid metabolism. Studies in animal models have demonstrated that increasing FGF21 activity can enhance metabolic function and promote longevity. Ongoing investigations into gene therapy to elevate FGF21 activity in humans consider it a promising therapeutic approach for the treatment of type 2 diabetes (DM2) and obesity. Gene therapy involving AAV-FGF21 has resulted in significant reductions in body weight, adipose tissue hypertrophy, inflammation, hepatic steatosis, inflammation, fibrosis, and insulin resistance for over one year in animal models subjected to a high-fat diet or ob/ob mice (obese mice) over extended periods. This therapeutic effect was achieved without side effects, despite maintaining elevated serum FGF21 levels persistently [143]. Additionally, mice overexpressing FGF21 exhibited a reduction in age-related thymus lipoatrophy [144].

However, there is evidence suggesting that circulating FGF21 levels increase with age in rodents and humans. The positive metabolic effects of FGF21 administration, known for promoting health, coincide with elevated hormone levels observed in obesity and diabetes. This increase may be attributed to altered tissue sensitivity to FGF21 [145]. Positive outcomes were observed in a gene therapy trial utilizing three different AAVs encoding FGF21, αKlotho, and the soluble form of mouse TGF-β receptor 2 (sTGFβR2) genes. The trial evaluated their effectiveness in mitigating the effects of four age-related diseases: obesity, DM2, heart failure, and kidney failure in animal models. Results included a 58% increase in cardiac function in mice with heart failure induced by ascending aortic constriction, a 38% decrease in α-smooth muscle actin expression, a 75% reduction in renal medullary atrophy in mice subjected to unilateral ureteral obstruction, and complete reversal of obesity and diabetes phenotypes in mice on a continuous high-fat diet [146].

### 5.5. Using Other Genes to Treat Age-Related Diseases

#### 5.5.1. Treatment of Age-Related Macular Degeneration

VEGF-A-mediated signaling is recognized as necessary and sufficient for the rejuvenation of a rapidly aging human organ, such as the skin, both at the morphological and molecular aging marker levels [86]. Furthermore, in mice treated with AAV-VEGF, mitochondrial dysfunction, metabolic disorders, endothelial cell senescence, and inflammation were reduced. This treatment resulted in increased longevity, decreased abdominal fat accumulation, reduced liver steatosis, mitigation of muscle mass loss (sarcopenia), bone mass loss (osteoporosis), decreased kyphosis, and a lower incidence of spontaneous tumors. These findings suggest that *VEGF* could be a promising target for anti-aging therapy [147]. However, certain age-related diseases require VEGF blocking, such as age-related macular degeneration of the yellow spot (AMD), a leading cause of blindness in the elderly. The gene therapy product ADVM-022 (AAV.7m8-aflibercept) is being developed for AMD treatment. A single intravenous injection of ADVM-022 has the potential to treat the wet form of AMD by providing sustained expression of therapeutic levels of intraocular anti-VEGF protein (aflibercept) and maintaining patients’ vision. ADVM-022 aims to reduce the ongoing treatment burden that often leads to treatment failure and vision loss in patients with wet AMD receiving anti-VEGF therapy in clinical practice (NCT03748784). Another clinical trial is underway in a similar direction to block VEGF signaling in AMD patients (NCT05657301).

A gene therapy option has been proposed using AAV1-BACE1 (encoding β-secretase) for injection into the pigment epithelium. In mice with a retinal macular degeneration phenotype (superoxide dismutase 2 (SOD2) knockdown mice), AAV1-BACE1 administration prevented the loss of retinal function and pathology, maintaining these effects for up to 6 months. Additionally, BACE1 overexpression inhibited oxidative stress, microglia changes, and loss of the integrity of tight contacts in the retinal pigment epithelium in mice [148]. Therapy with AAV-APNp1 (encoding adiponectin peptide 1) was also evaluated in mice with a model of AMD. The results showed increased APNp1 expression in the retina and vasculature over a 28-day period, which could be considered for further clinical trials [149]. Anti-inflammatory gene constructs, such as AAV-TatCARD, may also be considered for the treatment of inflammation in AMD and other posterior pole ocular diseases where inflammation may play a role [150]. The efficacy of gene therapy for AMD using subretinal delivery of AAV-NDI1-7 has been demonstrated. Additionally, this gene has shown utility in studies involving Parkinson’s disease, Leber’s hereditary optic neuropathy, and multiple sclerosis, reducing oxidative stress in these conditions [151].

#### 5.5.2. Cognitive Dysfunction/Neurodegenerative Diseases

A clinical trial (Phase 1) of gene therapy utilizing AAV2-BDNF (encoding brain-derived neurotrophic factor) to address early Alzheimer’s disease (AD) and mild cognitive impairment in 12 participants is currently ongoing. BDNF, a nervous system growth factor, regulates neuronal function in key memory circuits of the brain (entorhinal cortex and hippocampus), potentially enhancing cognitive function in patients [152] (NCT05040217). Astroglial expression of clusterin (*Clu*) mediated by AAV was found to promote excitatory neurotransmission in wild-type mice and restore synaptic deficits in Clu knockout mice. Overexpression of *Clu* in astrocytes of 5xFAD mice with a BA model resulted in reduced β-amyloid accumulation and complete restoration of synaptic deficits [153].

CERE-120, a gene product (AAV2 encoding the neurotrophic factor neurturin), has been developed to deliver a therapeutic factor to degenerating nigrostriatal neurons in Parkinson’s disease. While it has been shown to be safe, ongoing investigation in phase 2 clinical trials is examining its efficacy [154]. Additionally, PBFT02, an AAV1-based drug, is currently in clinical trials for gene therapy targeting frontal temporal dementia by delivering a functional copy of the human progranulin (*GRN*) gene into the brain. The study aims to assess the safety, tolerability, and efficacy of this treatment in patients with frontal temporal dementia and mutations in the *GRN* gene (NCT04747431).

Gene therapy has shown promise in improving the condition of animals with perioperative neurocognitive disorder. Downregulation of myeloid differentiation factor 2 (MD2) expression by AAV-shMD2 (encoding hairpin RNA) or injection of the synthetic MD2-disrupting peptide Tat-CIRP-CMA improved spatial reference learning ability and memory in anesthetized and surgically treated animals [155]. Finally, intracerebroventricular infusion of nerve growth factor (NGF) into the basal forebrain improved spatial memory in old animals. Alongside preventing age-related memory deficits, *NGF* gene transfer increased the size of cholinergic neurons by 34% in the medial septum. This approach may represent an effective therapy for age-related dementias associated with the dysfunction of cholinergic activity and memory, such as AD [156].

## 6. Conclusions

The increase in life expectancy is a positive trend in human history, but aging has become a prominent challenge that we must address using new technologies and precise tools. The emerging focus on senolytics, substances capable of removing senescent cells from the body, stems from successful clinical trials of some drugs. Senescent cells play a pivotal role in age-related diseases, and their elimination can offer significant therapeutic benefits. However, most senolytics currently exhibit serious side effects that need to be addressed for optimal results.

Cell therapy, involving the use of stem cells and specialized cells to repair damaged and aging tissues, holds promise for slowing the aging process and enhancing overall health. However, when using autologous stem cells from elderly patients, obtaining an unspecified therapeutic effect is necessary due to their intrinsic senescent nature.

The use of AAV to deliver genetic material and a gene that can slow down the aging process may be a revolutionary approach to treating aging, opening new horizons for slowing down and even reversing its processes. However, this is contingent on eliminating undesirable side effects and reducing production costs. While the studies described above have demonstrated inspiring success in improving the physical condition of model animals and patients, the centuries-long quest for a true “elixir of youth” to reverse age-related changes and eliminate diseases is far from over.

## Figures and Tables

**Figure 1 ijms-25-00643-f001:**
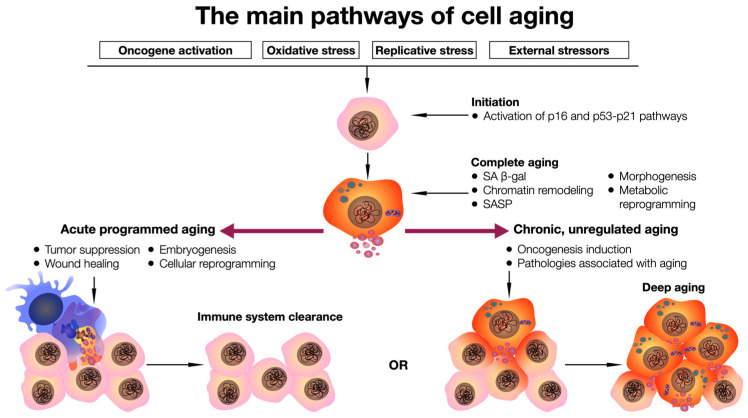
General scheme of the cell aging pathway. Exposure to stressors triggers activation of p16 and p53–p21 signaling pathways, which leads to complete cell aging, which can be completed by cell elimination from the tissue with the help of the immune system, or long-term preservation of a pathologically functioning cell and its pathological influence on the microenvironment in the tissue.

## Data Availability

Data available in a publicly accessible repository.

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
