# Peer review of "Eternal Youth: A Comprehensive Exploration of Gene, Cellular, and Pharmacological Anti-Aging Strategies"

_ijms, 2024, doi:10.3390/ijms25010643_

Round 1
Reviewer 1 Report
Comments and Suggestions for Authors
Review paper "Eternal Youth: A Comprehensive Exploration of Gene, Cellular, and Pharmacological Anti-aging Strategies" by K. V. Kitaeva, V. V. Solovyeva, N. L. Blatt and A. A. Rizvanov presents the latest findings from the field of aging research and the development of anti-aging therapeutic strategies.
The paper is well written and easy to read, covers the most important genetic and metabolic aspects of aging at the cellular and organismal level and lists the most important anti-aging drugs and therapeutic approaches with special reference to gene and cell therapy. It will certainly be a good source of information on this topic for scientists of a wider range of specialties.
Perhaps it should be pointed out that the original name of the p21 protein was SDI1 (Senescence Derived Inhibitor), which later received the p21WAF1/CIP1 versions to finally become just p21.
Also, in line 103 I think "cell formation." should be removed because it doesn't belong in this sentence.
The paper can be accepted after these minor corrections.
Author Response
Dear Reviewer,
We would like to express our gratitude for your thoughtful review of our paper titled "Eternal Youth: A Comprehensive Exploration of Gene, Cellular, and Pharmacological Anti-aging Strategies."
We have accepted and taken into account your comment about p21. Additionally, we acknowledge your observation in line 103, and we will promptly remove "cell formation." as it is indeed not relevant to the sentence. We are grateful for your positive feedback and constructive suggestions. We will make the necessary corrections promptly and resubmit the revised manuscript. Thank you once again for your time and valuable input. Sincerely yours, authors

Reviewer 2 Report
Comments and Suggestions for Authors
Major comments:
The manuscript requires an overall language evaluation.
The authors aim to explore the strategies and advancements in the field of anti-aging therapies; however, the objectives are not clearly defined. For instance, the advantages and disadvantages of using cell therapy depend on the target organ.
On the section “Cellular aging”, Do organs aged equally? Does the senescent phenotype present in a lung aged epithelium similar to the intestine epithelium? An evaluation of common aged hallmarks across organs and cell types versus specific to the disease could help to understand the lack of a universal strategy.
The focus of the cell therapy remains unclear. For instance, will it be used as an anti-aging strategy in areas where there is a high risk of cancer, heart disease or immune system dysfunction? Finally, the authors should provide a perspective on the review. What is it bringing to the audience?
From the conclusions, what would need to be accounted for in future studies that use cell therapy (or other) as an anti-aging strategy? What is the discussion provided by the authors? What is the rationale for this perspective?
Author Response
Dear Reviewer,
Thank you for your comprehensive review and valuable feedback on our manuscript titled "Eternal Youth: A Comprehensive Exploration of Gene, Cellular, and Pharmacological Anti-aging Strategies." We appreciate the time and effort you dedicated to evaluating our work.
Reviewer: The authors aim to explore the strategies and advancements in the field of anti-aging therapies; however, the objectives are not clearly defined. For instance, the advantages and disadvantages of using cell therapy depend on the target organ. On the section “Cellular aging”, Do organs aged equally? Does the senescent phenotype present in a lung aged epithelium similar to the intestine epithelium? An evaluation of common aged hallmarks across organs and cell types versus specific to the disease could help to understand the lack of a universal strategy.
Authors: We want to express our gratitude for your insightful comments regarding the clarity of our objectives and the need for further detail on specific aspects of our discussion. Based on your feedback, we have thoroughly revised the manuscript to address these concerns. Specifically, we have expanded the subchapter on cellular aging to include detailed information about the molecular genetic features of organ aging, encompassing organs such as the lungs and intestines. This addition aims to provide a more comprehensive understanding of the aging process at the cellular level, considering variations across different organs.
Reviewer: The manuscript requires an overall language evaluation.
Authors: We appreciated your attention to the language aspects of our manuscript. In response to your comments, we conducted a thorough language revision to ensure clarity, coherence, and adherence to academic writing standards.
Reviewer: The focus of the cell therapy remains unclear. For instance, will it be used as an anti-aging strategy in areas where there is a high risk of cancer, heart disease or immune system dysfunction? Finally, the authors should provide a perspective on the review. What is it bringing to the audience? From the conclusions, what would need to be accounted for in future studies that use cell therapy (or other) as an anti-aging strategy? What is the discussion provided by the authors? What is the rationale for this perspective?
Authors: A discussion regarding the potential of cell therapy has been incorporated into Section 4, titled "Cell and Genetically Modified Cell-Based Therapy." Furthermore, the perspective on gene and cell therapy has been elucidated in the "Conclusions" section.
We believe that these enhancements have significantly strengthened the manuscript, aligning it more closely with the objectives you outlined in your review.
Sincerely,
Authors
